

# Experimental warming decreases arbuscular mycorrhizal fungal colonization in prairie plants along a Mediterranean climate gradient

Hannah Wilson[1], Bart R. Johnson[2], Brendan Bohannan[3],
Laurel Pfeifer-Meister[3], Rebecca Mueller[1] and Scott D. Bridgham[3]

[1] Department of Biology, University of Oregon, Eugene, Oregon, United States
[2] Department of Landscape Architecture, University of Oregon, Eugene, Oregon, United States
[3] Institute of Ecology and Evolution, University of Oregon, Eugene, Oregon, United States

## ABSTRACT

**Background:** Arbuscular mycorrhizal fungi (AMF) provide numerous services to their plant symbionts. Understanding climate change effects on AMF, and the resulting plant responses, is crucial for predicting ecosystem responses at regional and global scales. We investigated how the effects of climate change on AMF-plant symbioses are mediated by soil water availability, soil nutrient availability, and vegetation dynamics.

**Methods:** We used a combination of a greenhouse experiment and a manipulative climate change experiment embedded within a Mediterranean climate gradient in the Pacific Northwest, USA to examine this question. Structural equation modeling (SEM) was used to determine the direct and indirect effects of experimental warming on AMF colonization.

**Results:** Warming directly decreased AMF colonization across plant species and across the climate gradient of the study region. Other positive and negative indirect effects of warming, mediated by soil water availability, soil nutrient availability, and vegetation dynamics, canceled each other out.

**Discussion:** A warming-induced decrease in AMF colonization would likely have substantial consequences for plant communities and ecosystem function. Moreover, predicted increases in more intense droughts and heavier rains for this region could shift the balance among indirect causal pathways, and either exacerbate or mitigate the negative, direct effect of increased temperature on AMF colonization.

Corresponding author
Hannah Wilson,
hwilson1@uoregon.edu

## INTRODUCTION

Arbuscular mycorrhizal fungi (AMF) are plant symbionts that colonize the roots of the majority of terrestrial plants; they provide enhanced nutrient and water uptake, increased drought and disease resistance, and increased plant productivity in exchange for carbon (C) (*Smith & Read, 2008*). AMF are a major contributor to terrestrial carbon and nutrient cycles (*Fitter, Heinemeyer & Staddon, 2000*) and are considered an important link between

above- and belowground processes (*Leake et al., 2004*). They can consume up to 20% of C produced by their plant host (*Bago, Pfeffer & Shachar-Hill, 2000*), and the hyphal network can occupy over 100 m $cm^{-3}$ of soil (*Miller, Jastrow & Reinhardt, 1995*), making up 20–30% of the total microbial biomass in terrestrial systems (*Leake et al., 2004*).

Given the widespread importance of AMF, it is not surprising that recent studies have concluded they may play a major role in mediating plant and ecosystem responses to climate change (*Rillig et al., 2002*, *Drigo, Kowalchuk & Van Veen, 2008*; *Compant, van der Heijden & Sessitsch, 2010*). The majority of studies have observed an increase in AMF colonization in response to experimentally increased $CO_2$ levels and/or temperature (*Compant, van der Heijden & Sessitsch, 2010*). However, many of these studies were performed with one or a few species of AMF and plant hosts under laboratory or greenhouse conditions (*Graham, Leonard & Menge, 1982*; *Baon, Smith & Alston, 1994*; *Staddon, Gregersen & Jakobsen, 2004*; *Heinemeyer et al., 2006*). Because AMF recently have been shown to have much higher species diversity than previously estimated (*Kivlin, Hawkes & Treseder, 2011*), and the benefits of AMF symbioses are not equal among plants (*Leake et al., 2004*), more studies are needed before generalizations can be made about the responses of AMF and their plant hosts to climate change.

A number of variables may influence AMF response to climate change. The general positive response of AMF colonization to increased $CO_2$ levels and temperature could be due to increased plant productivity, resulting in a larger demand for plant nutrients and enhanced production of root exudates (*Fitter, Heinemeyer & Staddon, 2000*; *Zavalloni et al., 2012*). Increased drought severity is a major concern for many regions, and AMF have been shown to enhance resistance to drought and improve water relations (*Augé, 2001*). However, a number of studies have found that increased drought can have a negative effect on AMF, depending on the species of AMF (*Davies et al., 2002*), hyphal growth within or outside the roots (*Staddon et al., 2003*), or the species of plant (*Ruiz-Lozano, Azcón & Gomez, 1995*). In a long-term climate manipulation, *Staddon et al. (2003)* found that increased AMF colonization in response to heat was mediated by soil moisture. Furthermore, they speculated that the effect of soil moisture could have been further mediated by changes in plant diversity and cover of various species, which were also highly correlated with mycorrhizal measures.

It is well established that a decrease in soil nutrient levels, especially of phosphorus (P) and nitrogen (N), can result in an increase in AMF colonization, whereas excess nutrients can result in lower colonization (*Mosse & Phillips, 1971*; *Smith & Read, 2008*). Thus, increased nutrient mineralization due to experimental warming could influence AMF growth (*Rillig et al., 2002*). Moreover, the ratio of N to P availability may also affect AMF responses to climate change (*Treseder & Allen, 2002*; *Johnson, 2009*). For example, *Blanke et al. (2012)* found that for plants grown in soils co-limited by N and P, P addition decreased colonization while N addition increased colonization.

To our knowledge, all previous experimental field studies of AMF-plant responses to climate change were performed at a single site. However, important factors such as soil characteristics and plant community composition often have high local variability.

To extrapolate site-specific results to a regional scale requires understanding the roles of both regional and local controls on AMF and plant responses. To this end, we used a manipulative climate change experiment embedded within a Mediterranean climate gradient in the Pacific Northwest (*Kottek et al., 2006*) to determine the underlying direct and indirect effects of increased temperatures on AMF and their plant hosts in Mediterranean climates. Mediterranean ecosystems contain a large percentage of global biodiversity of terrestrial plants (20%) in proportion to their total terrestrial area (5%) (*Cowling et al., 1996*). They are also among the most sensitive biomes to global climate change in terms of biodiversity (*Sala et al., 2000*).

We hypothesized that much of the effect of temperature on AMF colonization, as well as the host plants' nutrient composition and biomass, would be mediated through interactions with vegetation dynamics and the availability of soil water and nutrients. We were also interested in whether these effects were regionally consistent along a gradient of increasing summer drought stress.

## MATERIALS AND METHODS

### Site descriptions

We studied three prairie sites along a 520 km latitudinal climate gradient in the inland valleys of the Pacific Northwest (Table 1). The southernmost site is in southwestern Oregon near the town of Selma, the central site is in central-western Oregon near the city of Eugene, and the northernmost site is in central-western Washington near the town of Tenino. The sites occur along a gradient of increasing severity of Mediterranean climate from north to south (Table 1). The southern site has the most extreme seasonal variation, experiencing the wettest, coolest winters and driest, warmest summers. The central and northern sites have comparatively milder winters and summers in terms of rainfall and temperature, with the central site having warmer average summer and winter temperatures than the northern site. Global climate change models for the Pacific Northwest predict an increase in average annual temperatures of +3.0 °C by 2,080 (range +1.5 °C to over +5.8 °C) (*Mote & Salathé, 2010*). While average annual precipitation projections are highly variable among different emission scenarios and models (range −10 to +20% by 2,080), across models there is a consistent prediction of warmer, wetter winters (precipitation range +8 to +42%) and hotter, dryer summers (precipitation range −14 to −40%) (*Mote & Salathé, 2010*).

As is typical for a study spanning a large region, each site has a different soil type. The southern site is a loamy Mollisol (coarse-loamy, mixed, superactive, mesic Cumulic Haploxeroll), the central site is a silty-clay loam Mollisol (very-fine, smetitic, mesic Vertic Haploxeroll), and the northern site is a gravelly sandy loam Andisol (sandy-skeletal, amorphic-over-isotic, mesic Typic Melanoxerand). The southern site has a circumneutral pH, and the central and northern sites are mildly acidic (Table 1). These differences in soil characteristics translate into large differences in nutrient availability, with the southern site having much greater N and P availability (Fig. S1) and a greater N:P ratio (Table 1). The central site had moderately greater N and P availability and a lower N:P ratio than the northern site.

**Table 1 Site characteristics.** Climate data is from the PRISM model for the period 1971–2000 (http://www.prism.oregonstate.edu/).

| Site | Southern | Central | Northern |
|---|---|---|---|
| Latitude | 42°16′41″N | 44°01′34″N | 46°53′47″N |
| Longitude | 123°38′34″W | 123°10′56″W | 122°44′06″W |
| Elevation (m) | 394 | 165 | 134 |
| Mean precip. (mm) | 1,598 | 1,201 | 1,229 |
| Mean mon. temp. (°C) | 12.2 | 11.4 | 9.8 |
| Max. mon. temp. (°C) | 19.9 | 17.3 | 15.3 |
| Min. mon. temp. (°C) | 4.1 | 5.3 | 4.9 |
| Sand (%) | 31.4 | 36.4 | 73.9 |
| Clay (%) | 22.5 | 11.9 | 2.4 |
| Silt (%) | 46.0 | 51.6 | 23.7 |
| Total soil nitrogen (%) | 0.3 | 0.5 | 0.3 |
| Total soil carbon (%) | 3.4 | 7.3 | 4.9 |
| N:P ratio of soil | 5.4 | 1.1 | 1.7 |
| pH | 6.5 | 5.8 | 5.6 |

## Experimental design

In accordance with the predicted climate change for the Pacific Northwest (as stated above), we employed a fully factorial design of $+3\ °C$ above ambient canopy temperature, 20% increased precipitation, both increased temperature and precipitation, and ambient controls, with five replicate 3 m diameter plots of each treatment at each site. Heating treatments used infrared heaters (Kalglo heaters model HS 2420; Kalglo Electronics Co., Inc., Bethlehem, PA, USA) controlled by a dimmer system (*Kimball, 2005*; *Kimbell et al., 2008*). Dummy heaters were installed in non-heated plots to account for potential shading effects. The plots were isolated from the surrounding soil by burying an aluminum barrier to 40 cm depth, or to the depth of major obstruction. The precipitation treatments have resulted in only minor effects on all response variables for which it has been examined, including a wide array of plant responses (*Pfeifer-Meister et al., 2013*, and unpublished data). For this reason and series of other logistical constraints, we considered only the heated and control treatments in our present study.

Plots at each site were treated in 2009 with one or two applications of the herbicide glyphosate (spring and fall) followed by thatch removal and seeding with an identical mix of 33 annual (12 forbs, 1 grass) and perennial (15 forbs, 5 grasses) native prairie species within each plot. During the same period of the initial planting, we started the heating and precipitation treatments. For each site, we collected seed from the nearest local population of each species, or purchased seed from a native plant nursery that used first-generation plants from the nearest seed source. During the 2010 growing season, the most aggressive exotic species were weeded, but natural succession was allowed to occur afterwards resulting in a mix of species that were either intentionally seeded, came from the seed bank, or dispersed into the plots.

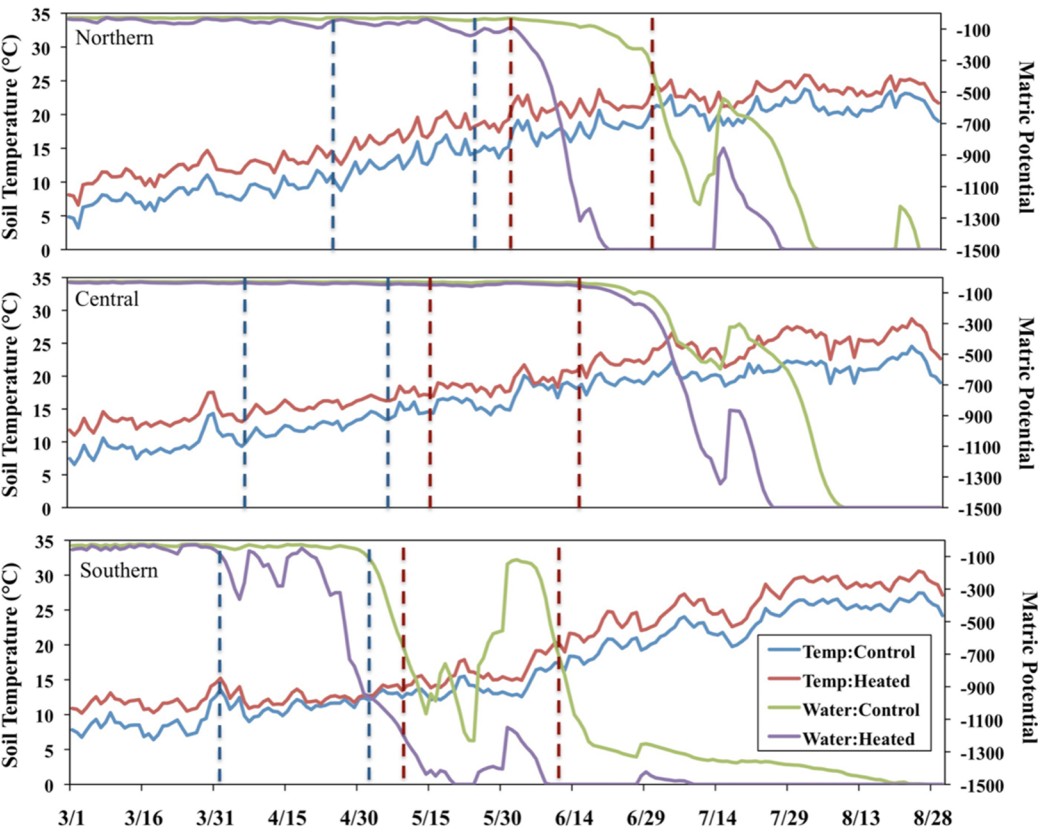

**Figure 1 Soil temperature and water availability in the 2011 growing season.** Panels correspond with sites, dotted lines indicate the time period used to estimate soil temperature and matric potential one month prior to plant collection. Because of the different phenology, red lines refer to perennial species and blue lines refer to the annual species.

In the 2011 growing season, we selected four native forbs for assessment of climate effects on AMF associations. Graminoid species were not assessed because no common species grew within all plots across all sites. The selected focal species were: *Achillea millefolium* L., Asteraceae (perennial); *Eriophyllum lanatum* (Pursh) Forbes, Asteraceae (perennial); *Plectritis congesta* (Lindl.) DC., Valerianaceae (annual); and *Prunella vulgaris* L. ssp. *lanceolata* (W. Bartram) Hultén., Lamiaceae (perennial).

## Plot measures

Soil temperature and volumetric water content were continuously monitored in the center of each plot with Campbell Scientific, Inc., Model 107 Temperature Probes and Campbell Scientific, Inc., CS616 Water Content Reflectometers, respectively. The average plot values for the one-month period prior to harvesting were used for analysis (Fig. 1). We considered other time frames, but this time period had the strongest correlation with AMF colonization. To enable comparison of soil water availability across sites, volumetric water content was converted to matric potential using site-specific values of soil texture and organic matter content (*Saxton & Rawls, 2006*).

Soil N and P availability were determined with anion and cation exchange probes (PNS) (Western Ag Innovations Inc., Saskatoon, Canada) that were inserted vertically 15 cm into the ground from April–July, 2011. $NH_4^+$-N and $NO_3^-$-N were combined into a single value for total inorganic N, although the value was dominated by $NO_3^-$-N.

Belowground net primary productivity (NPP) was measured using the root in-growth core method (*Lauenroth, 2000*) with 5 cm-diameter by 20 cm-depth cores. Aboveground NPP was estimated by destructive harvesting at peak standing biomass of a 0.30 $m^2$ area within each plot. All vegetation was dried to a constant mass at 60 °C before weighing. Aboveground biomass was also separated into forb and grass NPP. Total cover of all species was averaged per plot by using the point-intercept method (*Jonasson, 1983*) with two 1 $m^2$ quadrats of 25 points each. Presence/absence was determined for all species that were not hit by a pin in a plot, and they were assigned a cover of 0.4%. We calculated plant species diversity using the average of the two quadrats per plot using Simpson's Diversity Index (1/D).

## Individual plant measures

We harvested three individuals of each focal plant species within each heated and control plot. We had to limit the number of individuals and limit the harvest to one annual collection because the plants were relatively large, and uprooting them caused disturbance to the rest of the plots and other experiments. Plants were collected at peak flowering to maintain consistency in phenology across the treatments and sites; thus, the annual species was collected approximately one month before the perennial species (Fig. 1). We weighed aboveground plant material after drying at 60 °C for 48 h. Using subsamples of ground and dried material, we determined total P by performing a hydrogen peroxide-sulfuric acid digest (*Haynes, 1980*) using a Lachat BD-46 Digester (Hach Company, Loveland, CO, USA) and then measuring phosphate with the vanadate-molybdate colorimetric method (*Motsara & Roy, 2008*). Total C and N content were measured with a Costech Elemental Analyzer ECS 4010 (Costech Analytical Technologies Inc., Valencia, CA, USA).

Due to the small size of the annual species, *P. congesta*, we pooled individuals across plots within a treatment in order to obtain enough plant material to measure P at all of the sites, and N at the northern site, resulting in a sample size of one per treatment. Thus, we do not report pair-wise comparisons between treatments on plant P or N for this species for these sites.

## Mycorrhizal measures

The percentage of plant root colonized by arbuscular mycorrhizas (i.e., AMF colonization) is a measure of AMF abundance (*Vierheilig, Schweiger & Brundrett, 2005*). To quantify AMF colonization, a subsample of roots from each plant was taken and boiled in a 5% Sheaffer® black ink-to-white vinegar solution for 10 min after being cleared with 10% KOH (*Vierheilig et al., 1998*). Using the grid-intersect method (*McGonigle et al., 1990*), we calculated the percentage of arbuscules, vesicles, and total root colonization separately by counting the presence or absence of arbuscules and/or vesicles connected by characteristic AMF hyphae for each millimeter of root segment.

To compare the AMF community composition across treatments and sites, we performed an initial trial run with one of our focal plant species, *Eriophyllum lanatum*. DNA was extracted using the PowerPlant® DNA isolation kit, and purified with Zymo DNA Clean & Concentrator™. AMF DNA extracted from the plant roots was amplified by PCR using the AMF-specific rDNA primers AML2 (*Lee, Lee & Young, 2008*) and NS31 (*Simon, Lalonde & Bruns, 1992*). The resulting amplicons were sequenced using an Illumina HiSeq 2000 sequencer (Genomics Core Facility, University of Oregon). Preliminary sequence data were analyzed using MOTHUR (*Schloss et al., 2009*). Unfortunately, less than 1% (92/14,000) of the resulting sequences were identified as AMF using BLAST (the remaining were plant DNA sequences), and further community analyses were not performed. A list of species identified can be found in Table S1.

## Greenhouse study

Because of large differences in nutrient availability, pH, and texture among sites (Table 1), we performed a greenhouse experiment to determine the effect of soil type on AMF colonization. Ten previously germinated seedlings of each species were planted in flats containing soil outside the plots from each site. Plants grew for eight weeks in a climate-controlled greenhouse at a constant 25 °C under natural light (approximately 12–14 h a day) and were watered as needed to remain above wilting point. After eight weeks, we harvested all plants, measured the dry weight of aboveground biomass, and used the same protocol described above to quantify the AMF colonization for each plant.

## Data analysis of the greenhouse and field experiment (ANOVAs)

For the greenhouse experiment, we used two-way ANOVAs (species, soil type) to test for differences in AMF colonization and aboveground plant biomass. For the field experiment, we used three-way ANOVAs (species, site, and treatment) to test for differences in AMF colonization, aboveground plant biomass, soil N availability, soil P availability, the ratio of soil N:P availability, plant N content, plant P content, and the plant N:P ratio. Although we measured arbuscule, vesicle, and total colonization separately, arbuscule colonization never differed from total colonization, and vesicle colonization was minimal. Thus, we only report total colonization for both greenhouse and field experiments. Soil and plant nutrient analyses can be found in Tables S7–S12.

For all analyses of the greenhouse and field experiments, we performed separate ANOVAs on each species when there was a significant species interaction with any of the other main effects. Post-hoc comparisons were performed using Tukey's HSD. For both greenhouse and field data sets, we used an arcsine-square root transformation to normalize the AMF colonization data, and a logarithm transformation to normalize the plant biomass and nutrient data.

## Structural equation models of the field experiment

We used structural equation modeling (SEM) (*Grace, 2006*) to examine the effect of experimental warming on AMF colonization, and how it may be mediated by soil water availability, soil nutrient availability, and plot vegetation. We also assessed how interactions among these factors affected host plant nutrient content and biomass.

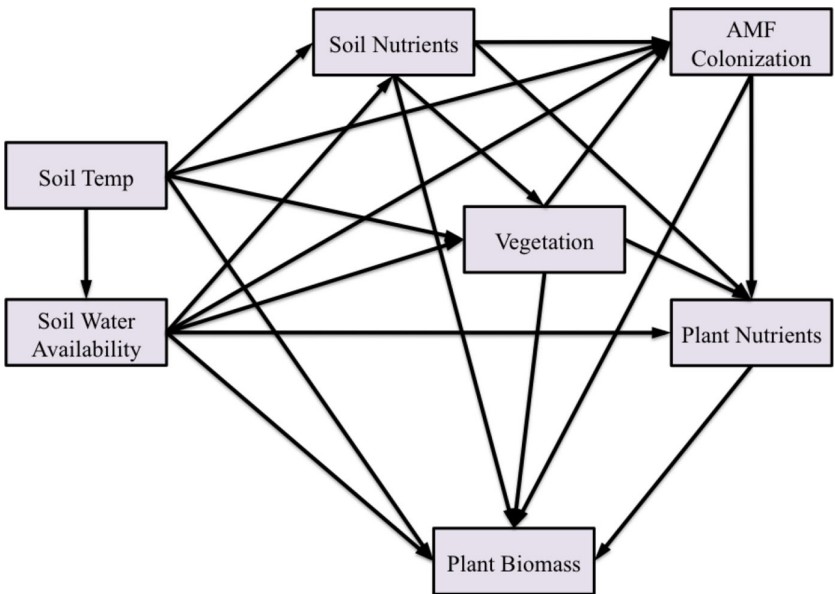

**Figure 2** **Structural equation model of the effect of temperature on AMF colonization and plant biomass.** We tested three a priori models that included either soil and plant nitrogen, soil and plant phosphorus, or soil and plant N:P ratios. Each box represents a variable in the model, while each arrow represents a predicted direct effect of one variable on another. A series of connected arrows through multiple variables represent indirect effects. The direct effect plus all the indirect effects of one variable on another is referred to as the total effect.

The greenhouse data suggested that soil type had a significant effect on AMF colonization, and we hypothesized this was due to differences in soil nutrient availability of P and/or N. However, the ratio of N:P has been suggested to be a more powerful predictor of AMF responses than availability of either nutrient alone (*Johnson, 2009*). Therefore, we developed three a priori SEMs to determine whether N, P, or the ratio of N:P had a larger effect in mediating AMF responses to temperature. Each model was identical except for the specific variables used to represent soil nutrients or plant nutrients in Fig. 2.

NPP and plant species diversity have been shown to affect AMF (*Vandenkoornhuyse et al., 2003*; *Johnson et al., 2004*). We tested our models using above- and belowground NPP, the ratio of above:below NPP, grass and forb NPP, the ratio of grass:forb NPP, and plant species diversity, using each separately as the vegetation variable in Fig. 2. Plant diversity had the greatest effect on AMF colonization, so we dropped the NPP measures from subsequent analyses to simplify our models.

The maximum likelihood method was used for model evaluation and to estimate the standardized path coefficients (*Grace, 2006*). For all analyses, we present only models that had good model fit as estimated by Pearson's chi-square goodness of fit ($\chi^2$) ($P > 0.05$ indicates good model fit), the Bentler Comparative Fit Index (CFI) ($< 0.90$ indicates good model fit), and the Root Mean Square Error of Approximation (RMSEA) ($< 0.05$ indicates good model fit) (*Bentler, 1990*; *Grace, 2006*). For models with good fit, we present only path coefficients that were significant at $P < 0.10$. All SEM analyses were performed using Amos 20.0 SEM software (SPSS Inc., Chicago IL, USA).

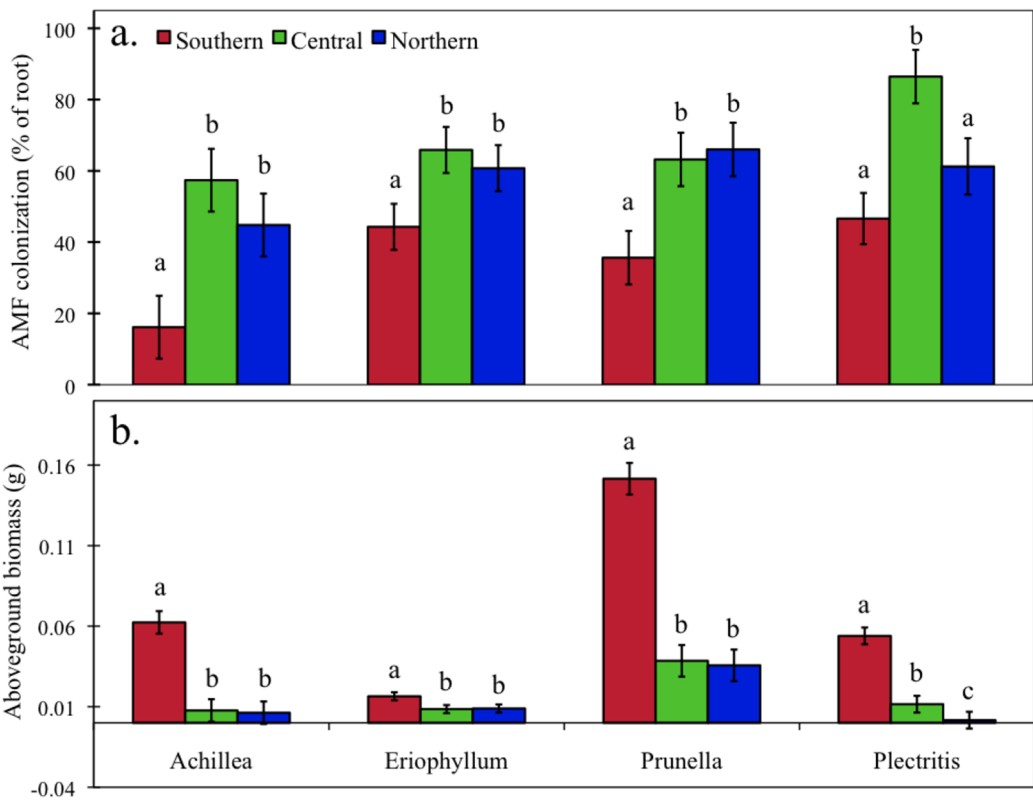

**Figure 3 Greenhouse experiment.** AMF colonization (A) and aboveground biomass (B) of the four species grown in soil from the three sites (Southern, Central, Northern) in a greenhouse. Different letters indicate significant differences among sites within a species. Error bars represent +/− one SE.

## RESULTS

### Greenhouse experiment

AMF colonization differed in plants grown in the three soils [$F(2, 103) = 37.4$, $P < 0.0001$] and among the four species [$F(3, 103) = 11.7$, $P < 0.001$], and the effect of soil type marginally depended on species [$F(6, 103) = 1.9$, $P = 0.09$, Fig. 3A]. For the three perennial species, we consistently found that plants grown in soil from the southern site had the lowest colonization ($P < 0.001$), whereas plants grown in soil from the central and northern site did not differ. The annual species, *P. congesta*, had the greatest colonization when grown in soil from the central site ($P = 0.006$).

Aboveground plant biomass differed by soil type [$F(2, 103) = 187.6$, $P < 0.001$] and among the four species [$F(3, 103) = 97.2$, $P < 0.001$], and the effect of soil type depended on species [$F(6, 103) = 23.9$, $P < 0.001$]. Despite the significant interaction, we found a consistent trend among the three perennial species, which were largest when grown in soil from the southern site ($P < 0.001$, Fig. 3B), whereas plants grown in the central and northern site soil did not differ in size. The annual species, *P. congesta*, was largest when grown in southern site soil, intermediate in the central site soil, and smallest in the northern site soil ($P < 0.000$).

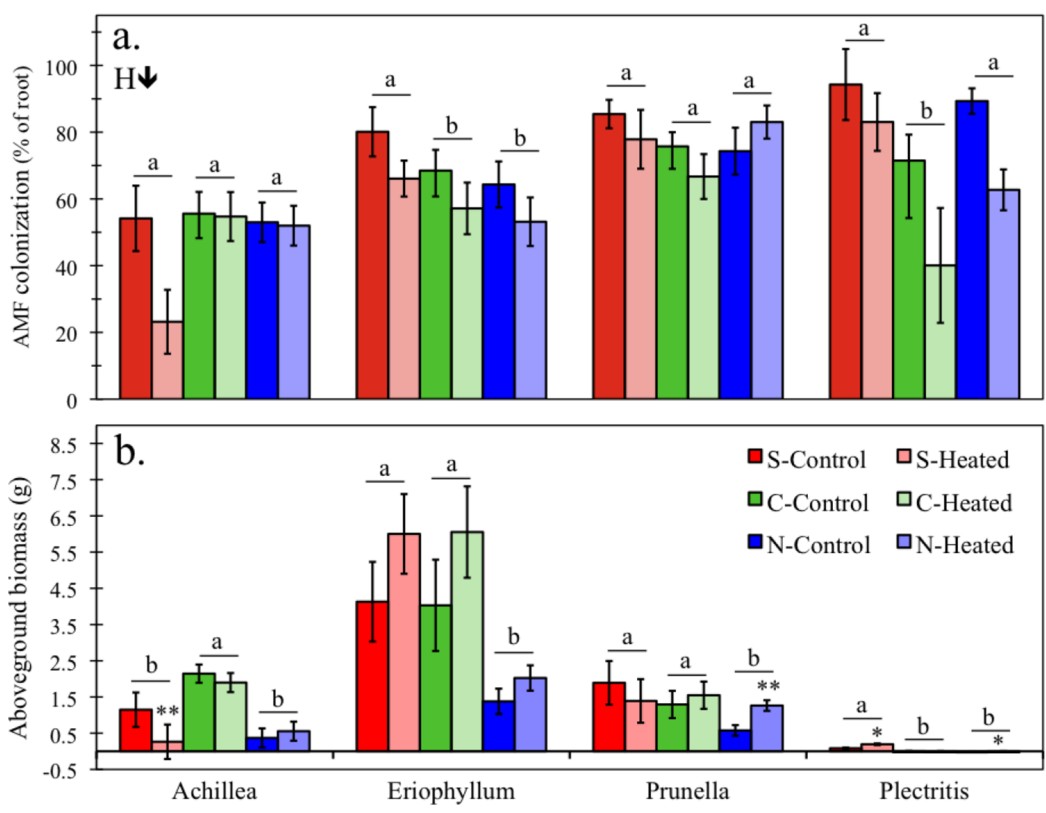

**Figure 4 The effect of heating on AMF colonization (A) and aboveground plant biomass (B) of the four plant species collected from the three sites (S = southern, C = central, N = northern).** H↓ represents a significant inhibitory main effect of heating. Different letters indicate significant differences among sites within a species. Asterisks represent significant differences between control and heated treatments (** = $P < 0.01$, * = $P < 0.10$). Error bars represent +/− one SE.

### Field experiment

Differences in AMF colonization among sites depended on species [$F_{(6, 281)} = 4.4$, $P < 0.001$, Table S2]. Colonization did not differ among the sites for *A. millefolium* and *P. vulgaris*, but *E. lanatum* marginally had the greatest colonization in the southern site ($P < 0.07$), and *P. congesta* had the lowest colonization in the central site ($P = 0.01$). Across all sites and species, heating consistently lowered colonization [$F_{(1, 281)} = 17.8$, $P < 0.001$, Fig. 4A].

Differences in aboveground plant biomass among sites depended on species [$F_{(6, 290)} = 19.6$, $P < 0.001$, Table S3]. *A. millefolium* was largest at the central site ($P < 0.001$) and *E. lanatum* and *P. vulgaris* were smallest at the northern site ($P \leq 0.05$). *P. congesta* was largest at the southern site ($P < 0.001$).

The effect of the heating treatment on plant biomass depended on both site [$F_{(2, 290)} = 4.6$, $P = 0.01$] and species [$F_{(3, 290)} = 3.5$, $P = 0.02$]. Heating decreased the size of *A. millefolium* plants at the southern site ($P = 0.001$), increased the size of *P. vulgaris* plants at the northern site ($P = 0.001$), and increased the size of *P. congesta* plants in the southern and northern sites ($P \leq 0.042$). *E. lanatum* size was not affected by heating treatments, although it trended toward larger plants in the heating treatments across sites (Fig. 4B).

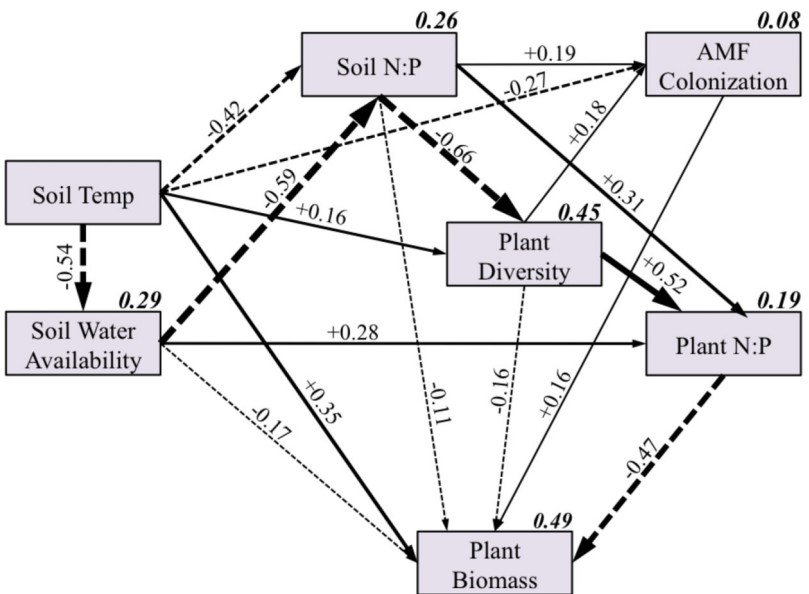

**Figure 5 Overall structural equation model including all sites and species.** Each box represents a variable in the model, while the number above each arrow represents the value of the standardized path coefficients. The width of each arrow corresponds with the magnitude of the path coefficient, solid lines indicate positive effects, and dashed lines indicate negative effects. Path coefficients not significant at $P < 0.10$ are not shown. The italicized, bold number above each box represents the total explained variance ($R^2$) of each variable.

## Structural equation models

To test the three a priori SEMs (Fig. 2), we used data across all sites and species (for a table of means and Pearson's correlations of the data, see Tables S4 and S5). Both the P and the N:P ratio model had good model fit (P SEM: $\chi^2 = 0.081$, $P = 0.78$, $CFI = 1.0$, $RMSEA < 0.0001$; N:P ratio SEM: $\chi^2 = 0.002$, $P = 0.96$, $CFI = 1.0$, $RMSEA < 0.0001$), while the N model had poor model fit ($\chi^2 = 26.3$, $P < 0.0001$, $CFI = 0.96$, $RMSEA = 0.284$) and was dropped from further consideration. Both the P and N:P ratio models had similar magnitudes and directions of the path coefficients. However, the plant N:P ratios suggested N limitation or N and P co-limitation (Fig. S2C), as plants with a ratio < 10 and > 20 are considered to be N limited and P limited, respectfully (*Güsewell, 2004*). Thus, we chose the N:P model (Fig. 5) for further interpretation (see Fig. S3 for P only model results).

We also examined the consistency of the N:P SEM model among each species and each site separately. Models for individual species showed similar patterns as the model using all species, but had poor model fit, presumably due to the lower sample size (N < 100), and we do not consider them further. Similarly, models that included data from each site separately had poor model fit, except for the model that included data from only the southern site ($\chi^2 = 0.76$, $P = 0.38$, $CFI = 1.0$, $RMSEA < 0.0001$). We were particularly interested in the SEM of the southern site because this site has much higher nutrient availability (Table 1), and the heated plots were beginning to experience extreme drought conditions at the time of plant collection (Fig. 1).

**Table 2 Standardized direct, indirect, and total effects of the overall N:P ratio SEM.**

| Effect of variable 1 | On | Variable 2 | Direct effect | Indirect effect | Total effect |
|---|---|---|---|---|---|
| Soil temperature | → | Soil water availability | −0.54 | N/A | −0.54 |
| Soil temperature | → | Soil N:P | −0.42 | 0.32 | −0.10 |
| Soil temperature | → | Plant diversity | 0.16 | 0.00 | 0.26 |
| Soil temperature | → | AMF colonization | −0.27 | 0.04 | −0.23 |
| Soil temperature | → | Plant N:P | N/A | −0.02 | −0.02 |
| Soil temperature | → | Plant biomass | 0.35 | 0.04 | 0.39 |
| Soil water availability | → | Soil N:P | −0.59 | N/A | −0.59 |
| Soil water availability | → | Plant diversity | −0.08 | 0.39 | 0.31 |
| Soil water availability | → | AMF colonization | −0.03 | −0.05 | −0.08 |
| Soil water availability | → | Plant N:P | 0.26 | −0.02 | 0.24 |
| Soil water availability | → | Plant biomass | −0.17 | −0.11 | −0.28 |
| Soil N:P | → | Plant diversity | −0.66 | N/A | −0.66 |
| Soil N:P | → | AMF colonization | 0.19 | −0.12 | 0.07 |
| Soil N:P | → | Plant N:P | 0.31 | −0.34 | −0.03 |
| Soil N:P | → | Plant biomass | −0.11 | 0.13 | 0.02 |
| Plant diversity | → | AMF colonization | 0.18 | N/A | 0.18 |
| Plant diversity | → | Plant N:P | 0.52 | −0.01 | 0.51 |
| Plant diversity | → | Plant biomass | −0.16 | −0.21 | −0.37 |
| AMF colonization | → | Plant N:P | −0.06 | N/A | −0.06 |
| AMF colonization | → | Plant biomass | 0.15 | 0.03 | 0.18 |
| Plant N:P | → | Plant biomass | −0.47 | N/A | −0.47 |

The overall SEM was fairly successful in explaining the variance in soil N:P ($r^2 = 0.26$), plant diversity ($r^2 = 0.45$), and plant biomass ($r^2 = 0.49$), but less successful in explaining the plant N:P ratio ($r^2 = 0.19$) and AMF abundance ($r^2 = 0.08$). Although all but three predicted path coefficients from the a priori model (Fig. 2) were significant (Fig. 5), we focus only on the direct and indirect effects on AMF colonization and plant biomass to simplify our presentation.

While temperature had a moderately strong direct negative effect on AMF colonization, as we hypothesized (Fig. 2), there were also many indirect effects of temperature on AMF colonization that were mediated by soil water availability, soil N:P, and plant diversity (Fig. 5). Soil N:P and plant diversity had moderate direct positive effects on AMF colonization. Soil water availability did not have a significant direct effect on AMF colonization, although it did have considerable indirect effects which were mediated by both soil N:P and plant diversity. Because some indirect pathways were positive and some were negative, the total indirect effect of temperature on AMF colonization as mediated by other variables was negligible (Table 2). Thus, the total effect of temperature on AMF colonization was predominately the direct negative effect.

Similarly, there were many indirect effects of temperature on the host plant biomass, which were mediated by soil water availability, soil N:P, plant diversity, AMF colonization, and plant N:P ratio. However, similar to AMF colonization, the various negative and

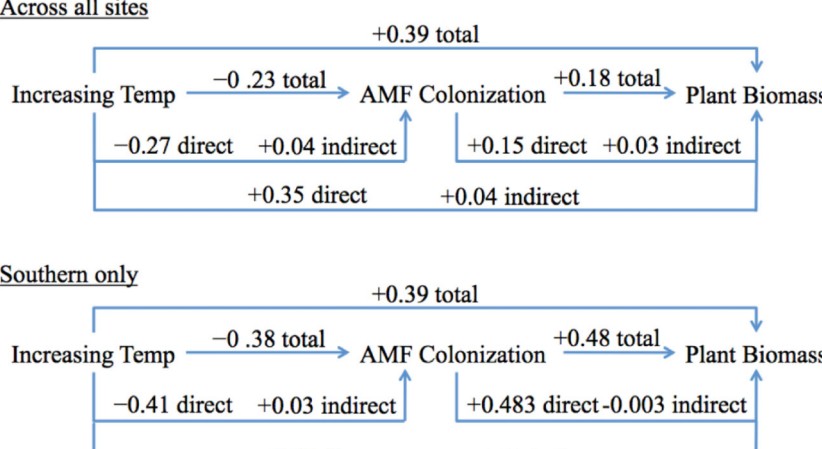

**Figure 6 Simplified scheme of direct, indirect and total effects of temperature on AMF colonization and plant biomass for the overall SEM and southern only SEM.** Total effect is the sum of direct and indirect effects. Numbers used are extracted from Tables 2 and S6, respectively.

positive indirect effects canceled each other out, and the total effect of temperature on plant biomass was largely a direct effect (Table 2). AMF colonization had a modest positive effect on plant biomass, which was also driven by direct, rather than indirect effects (Table 2; Fig. 6). Contrary to our expectations, AMF colonization did not affect plant N:P ratios, though plant N:P ratios had a strong negative effect on plant biomass.

Even though the southern site SEM had fewer significant pathways than the overall SEM, and the path coefficients were different in magnitude (and occasionally direction), the general outcomes were very similar to the overall SEM (Figs. 6 and S4). The effect of temperature on AMF colonization was still largely a direct negative effect, as indirect effects canceled out. The effect of temperature on plant biomass was also predominately a positive direct effect (Fig. 6; Table S6).

The southern site SEM was different in that there was a stronger negative effect of temperature on AMF colonization, and AMF colonization had a much stronger positive effect on plant biomass. The total explained variance of AMF colonization was higher for the southern site than the overall SEM (19% compared to 8%; Figs. S4 and 5, respectively). The total effect of temperature on plant biomass was, however, identical to the overall SEM (0.39, Fig. 6), and the total explained variance in plant biomass was very similar (44% compared to 49%, Figs. S4 and 5, respectively).

## DISCUSSION

### Site-level effects: comparing greenhouse and field experiments

In the greenhouse experiment, which was used to isolate the effects of soil type on AMF colonization and host plant response, we found the expected pattern of higher colonization in the soils with lower nutrient availability (*Mosse & Phillips, 1971*; *Smith & Read, 2008*). Even though colonization was higher in the central and northern site soils,

plants were consistently smaller, suggesting that increased colonization did not fully compensate for the large differences in soil nutrient availability between the southern site and the other two sites.

In contrast, in the field experiment we saw few overall differences in AMF colonization among sites. Although there was a general trend for plants at the northern site to be smaller, differences in plant size between the southern and central site were not consistent among species. These results suggest that the effects of climate may overwhelm the effect of soil type and nutrient availability on AMF colonization and plant biomass, although we cannot exclude the possibility that other site-level factors were important.

## Heating effects

The most intriguing result from the field experiment was the consistent decrease in AMF colonization in the heating treatment, in contrast to the positive effects reported from the majority of similar warming studies (*Compant, van der Heijden & Sessitsch, 2010*). There also was a general trend of increased aboveground biomass in the heating treatments, which is consistent with treatment effects on 12 different species from a related experiment that used the same climate manipulation (*Pfeifer-Meister et al., 2013*) and aboveground NPP collected at the plot level (data not shown).

We used SEM to test our hypothesis that the effect of temperature on AMF and their plant hosts would be mediated by indirect interactions with soil water availability, soil nutrients, and plant species diversity. However, because of both negative and positive interactions, these indirect effects canceled out, and the total effect of temperature was driven by the direct effects (Fig. 6; Table 2). We also demonstrated that this result was regionally consistent across the Mediterranean climate gradient represented by our three sites, despite the local site effects of soil type demonstrated in the greenhouse experiment. We are confident this result was not driven primarily by innate site differences because the southern site SEM had similar effects of temperature, despite some differences in causal pathways (Fig. S4). Moreover, the ANOVA results from the field experiment support the finding of a negative heating effect across sites and species (Fig. 5).

However, our analysis was limited to a single growing season after less than two years of heating. Over time, the effect of increasing temperature could make these indirect effects stronger or alter the balance among them. Additionally, 2011 was a La Niña year with greater spring precipitation than in other years. Moreover, the Pacific Northwest and Mediterranean regions globally are predicted to experience increasingly severe summer drought and heavier winter rains over the 21st century (*Mote & Salathé, 2010*; *Ruffault et al., 2012*). Thus, indirect effects mediated by soil water moisture could become more prominent in the future. Given these considerations, we examine the direct and indirect pathways in some detail below.

## Indirect effects

Deconstructing the total effects into indirect and direct effects helped reveal possible mechanisms that could be responsible for the results we found, and we discuss a few examples of these complicated indirect effects as follows.

The total effect of temperature on soil N:P was nearly neutral because the direct effect was negative (−0.42) and the indirect effect was positive (+0.32) (Table 2; Fig. 5). It seems, however, that the negative direct effect was driven by innate site difference in soil type (not the heating treatments). This negative relationship was clearly shown in a scatter plot of soil N:P vs. temperature (data not shown), where the soil N:P ratio was much higher in the southern site than the more northern sites, but temperatures during this time period were higher in the northern site than southern site (Table S4).

The positive indirect effect of soil temperature on soil N:P was driven by the negative effect of soil temperature on soil water availability, which in turn had a strong negative effect on soil N:P (resulting in a net positive effect). This positive effect agrees with our nutrient data (Table 1), where we saw an increase in soil N:P in the heating treatments in two of the three sites. Additionally, in the southern only SEM there was only a positive effect of soil temperature on soil N:P (Fig. S4).

Assuming that the negative direct effect of soil temperature on soil N:P was mainly driven by innate differences in soil type among the sites, our results suggest that increasing soil temperatures caused a shift toward P limitation due to a decrease in soil water availability. This may reflect the much greater mobility of nitrate (the predominant form of inorganic N in our sites) than P in soils. Increasing soil N:P had a moderate direct positive effect on AMF colonization, and it has been shown that plants in P-limited soils tend to have increased colonization and produce more exudates known to attract AMF (*Ostertag, 2001*; *Yoneyama et al., 2012*). The positive effect of warming on AMF colonization that most other studies have found could have been due to increased P limitation mediated by soil water availability (*Rillig et al., 2002*; *Staddon et al., 2003*). The relative limitation of P and N has been previously suggested as an important driver of AMF responses (*Johnson, 2009*). Testing all three of the a priori SEMs revealed that the N:P ratio was a better predictor of AMF colonization than the availability of soil N or P alone.

Although it makes sense that increasing soil P limitation would increase AMF colonization, the positive direct effect (+0.19) was diminished by the negative indirect effect (−0.12) mediated via plant diversity. Consistent with previous studies of the effect of plant diversity on AMF (*Vandenkoornhuyse et al., 2003*; *Johnson et al., 2004*), plant diversity had a positive effect on AMF colonization (+0.18). Because increasing soil N:P had a strong negative effect on plant diversity (−0.66), this indirect effect of soil N:P on AMF colonization was negative.

We found that plant species diversity was a better predictor of AMF colonization than various measures of net primary productivity (see *Plot Measures*). While it has been suggested that increased productively should directly affect AMF by increasing belowground C allocation (*Pendall et al., 2004*), it has also been shown that nutrient and C allocation are not shared equally among the plant and fungal symbionts within a community (*Klironomos, 2003*; *van der Heijden, Wiemken & Sanders, 2003*; *Leake et al., 2004*). Higher plant diversity may provide an improved root network that accommodates both higher colonization and AMF diversity (*van der Heijden, Wiemken & Sanders, 2003*; *Leake et al., 2004*).

### Direct effects

The direct negative effect of temperature on AMF colonization could have been a physiological response of the AMF (*Koltai & Kapulnik, 2010*). However, the total explained variance in AMF colonization for both the overall and southern only SEM was small (8 and 18%, respectively), and the direct effect may have been mediated by something we did not measure. Increased temperatures have been shown to decrease extraradical hyphae, presumably due to higher decomposition and turnover rates (*Rillig et al., 2002*; *Rillig, 2004*; *Wilson et al., 2009*). Because extraradical and internal root colonization has repeatedly been shown to be positively correlated (*Wilson et al., 2009*; *Barto et al., 2010*; *van Diepen et al., 2010*), we predict we would have observed a decrease in extraradical hyphae as well, had it been measured. A decrease in extraradical hyphae drastically decreases glomalin production, a glycoprotein that has been shown to increase soil stability (*Rillig, 2004*). Decreased AMF colonization could have serious consequences to overall ecosystem functions by destabilizing soil aggregates (*Wilson et al., 2009*).

We saw a positive total effect of temperature on plant biomass in both the overall and southern SEM, which was also primarily driven by the direct effect. Likewise, we found a modest positive total effect, driven primarily from the direct effect, of AMF colonization on plant biomass in the overall SEM (Fig. 6). The same was true for the southern site SEM, but the effect of AMF colonization on plant biomass was much stronger (Fig. S4). Although the total effect of heating on biomass was positive, over time the indirect negative effect on plant biomass (via the negative temperature effect on AMF colonization) could dampen the total positive effect of temperature on plant biomass, in addition to other ecosystem consequences.

### AMF community data

AMF colonization did not have any effect on plant N:P ratios (Figs. 5 and S4) or plant P content (Fig. S3). Although AMF are well known for enhancing P uptake, it has been shown that enhanced uptake via the AMF symbiont is not necessarily correlated with the degree of AMF colonization or the P content in the plant (*Smith, Smith & Jakobsen, 2004*). However, plant species diversity had a relatively strong effect on plant N:P (negative effect in the southern-only SEM and positive effect in the overall SEM), which could have been mediated by the community of AMF, rather than the overall colonization (*van der Heijden et al., 1998*; *Klironomos et al., 2000*; *van der Heijden, Wiemken & Sanders, 2003*).

Although our community data are limited, we do have evidence that there was a diverse community of AMF across and within the sites. Our community data set (from one host plant species, *E. lanatum*) spans most major families of the Glomeromycota (Fig. S5; Table S1). It would be interesting to further investigate the links between plant species diversity, AMF community, and plant nutrient uptake under climate change.

## CONCLUSIONS

We found that the direct effect of increasing temperatures caused a decrease in AMF colonization, and this appeared to be regionally consistent across the Mediterranean

climate gradient. A suite of complicated indirect effects mediated this response, although these effects canceled out due to both positive and negative effects. However, because of the fine balance of indirect effects, this region could potentially be quite sensitive to climate change. Over time, a shift in the relative strengths of different indirect effects could either exacerbate or mitigate the negative direct effect of temperature on AMF colonization. Furthermore, we cannot rule out the possibility that the direct effect may have been mediated by other variables we did not measure, such as glomalin secretion and related effects on soil stability. AMF colonization appears to be most important for plant biomass production in the southern site, the most extreme site in terms of Mediterranean seasonality. Thus, should ecosystems in Mediterranean climates experience even more intense droughts and heavier rains as predicted under many climate change scenarios, a subsequent decrease in AMF colonization could have substantial consequences for plant communities and ecosystem function.

To our knowledge, this is the first manipulative climate change study to examine the regional response of AMF interactions. Interestingly, our results challenge the conventional view that AMF respond positively to increased temperature. Many previous studies, however, were either performed in a greenhouse or at a single site, potentially limiting the generality of their results. Our research highlights how multi-site experiments at the regional level are needed to make reliable generalizations about the response of AMF-plant interactions to climate change.

## ACKNOWLEDGEMENTS

Much appreciation goes out to The Nature Conservancy, the Center for Natural Lands Management, and the Siskiyou Field Institute for providing the location of the field sites. Special thanks to Maya Goklany for her help with the analysis, Dr. Timothy Tomaszewski and Lorien Reynolds for providing essential data, and Roo Vandergrift for assistance with laboratory work.

### Funding
This research was funded by the Office of Biological and Environmental Research, Department of Energy (DE-FG02-09ER604719). The funders had no role in study design, data collection and analysis, decision to publish, or preparation of the manuscript.

### Grant Disclosures
The following grant information was disclosed by the authors:
Office of Biological and Environmental Research, Department of Energy: DE-FG02-09ER604719.

### Competing Interests
The authors declare that they have no competing interests.

## Author Contributions

- Hannah Wilson conceived and designed the experiments, performed the experiments, analyzed the data, contributed reagents/materials/analysis tools, wrote the paper, prepared figures and/or tables, reviewed drafts of the paper.
- Bart R. Johnson conceived and designed the experiments, reviewed drafts of the paper, general advising.
- Brendan Bohannan contributed reagents/materials/analysis tools, reviewed drafts of the paper, general advising.
- Laurel Pfeifer-Meister conceived and designed the experiments, reviewed drafts of the paper, general advising.
- Rebecca Mueller contributed reagents/materials/analysis tools, reviewed drafts of the paper, guidance on lab techniques.
- Scott D. Bridgham conceived and designed the experiments, analyzed the data, contributed reagents/materials/analysis tools, reviewed drafts of the paper, general advising.

## Data Deposition

The raw data has been supplied as Supplemental Dataset Files.

## Supplemental Information

Supplemental information for this article can be found online at http://dx.doi.org/10.7717/peerj.2083#supplemental-information.

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
