# Peer review of "Experimental warming decreases arbuscular mycorrhizal fungal colonization in prairie plants along a Mediterranean climate gradient"

_PeerJ, doi:10.7717/peerj.2083_

## Round 0.1 · original submission · Major Revisions

In general the reviewers agree about the relevance and strength of the study. However as indicated by one of the reviewers, some methodological and experimental design clarifications need to be addressed to ensure that this study can be replicated if needed in the future. I would encourage the authors to address these and all other comments brought up by all reviewers.

Reviewer 1 ·

Basic reporting

No Comments

Experimental design

No Comments

Validity of the findings

No Comments

Additional comments

This is an interesting paper and has potential for publication.

·

Basic reporting

Title: suggest a title change to reflect the experimental design
Introduction: No mention of the role of fire in the system. Need to articulate ecological "drivers" of GCC (Tylianakis et al. 2008). No background to precipitation pattern. In addition to spatial variation, there is also temporal variation which differentiates the trajectory of change and susceptibility to change.
Methods: there are some serious deficiencies in the methods section, in that, the experiment could not be repeated as there is in sufficient information about how the precipitation and heating treatments were carried. there is no explanation of why 20% increase of precipitation was chosen, or how the treatment was calculated or applied (did they use rainwater?).
Results: it was not clear whether the soil heating treatment worked? only mycorrhizal species from one host plant is presented. what about the other three? no linkage between species diversity of symbiont and change in colonization pattern was identified.
Discussion: the discussion starts of ok, but then there are some extensions which are beyond the scope of the study, and extrapolations as to how these apply to the present study and not justified.

Experimental design

there are serious deficiencies in the description of the experimental approach, i.e. we could not repeat experiment with the information provided. the heating treatment was targeting canopy and soil, yet only soil data collected. how was the precipitation study executed.? how long was the duration of the treatments?

Validity of the findings

the lack of mention of the why the precipitation study failed to elicit a response needed to be discussed rather than ignored in relation to the author's primary hypothesis around "mediated by vegetation dynamics and availability of soil water and nutrients".
I have attached a modified MSword document with direct comments.

Additional comments

There are some details which are lacking in five key areas that are crucial to the study:
1. Average annual precipitation of the region, nature of the precipitation study, and justification of 20% increase.
2. Length of time of the treatments, evidence of canopy / soil heat treatment effect.
3. Explanation of why precipitation has no effect on soil water availability?
4. Fungal symbionts in the other three test species, and discussion of species diversity of AMF fungi in relation to colonization rates.
5. How was role of "vegetation dynamics" assessed / tested?
While the primary hypothesis is well articulated, there are some instances where the discussion extends beyond the scope of the data, and does address the whether the primary hypothesis is supported or rejected.

Reviewer 3 ·

Basic reporting

Wilson et al have described patterns of arbuscular mycorrhizal fungal (AMF) colonization of 4 forb species from Mediterranean habitats in response to artificial warming at three sites across a temperature and precipitation cline in the Pacific Northwest of North America. This is a massive experiment that aims to how the effect of warming on the mycorrhizal association affects plant performance by utilizing structural equation modeling (SEM). Across all sites artificial warming directly decreases the percentage of root length colonized by mycorrhizal fungi, and has a range of positive and negative indirect effects on plant growth that would lead to site (soil properties such as fertility), year (ie, precipitation level variation) and species specific variation in cumulative effects. However, in total I feel this is a useful and important contribution to our understanding of how soil warming will impact a critical symbiosis that underlies the productivity and species diversity of most terrestrial ecosystems.
The submission adheres to PeerJ guidelines (as far as I have looked), is in clearly written English, has sufficient background for a non-specialist should be able to follow it, has figures that are clear in the submitted pdf, is self-contained, and appears to have made raw data available.

Experimental design

The design that was employed is ambitious and provides much of the power of the experiment. At three Mediterranean grassland sites along a North-South cline in the Pacific Northwest artificial warming plots were installed, along with a range of control plots. All pre-existing vegetation was removed by herbicide, and a common community was planted from seeds collected from nearby populations of native plant species or from nurseries. This sets the stage for nearly identical communities at 3 distinct sites with different treatments. A possible drawback is that AMF communities may have differed based on prior plant communities; however, there is no clear solution I can see to this risk, and I am uncertain it would have a large effect anyway. After two years, the four species that persisted in all treatments and sites were harvested in a targeted fashion to estimate root colonization by AMF, aboveground biomass, and other community estimates such as productivity (aboveground biomass and root biomass of the total plant community) and plant species diversity. And attempt was also made to identify AMF by next generation sequencing. This approach mostly failed and is only mentioned, as mostly plant DNA was amplified and few fungi could be identified per Illumina read. A companion greenhouse experiment was performed to look at the four species in isolation, without the complication of co-occuring plant taxa and the warming treatments.
The meat of the results is a structural equation model of the data, using a basic model that predicts the relationships of soil temperature, moisture, fertility, AMF colonization, vegetation, and plant nutrients on plant biomass. I think the group has the right basic relationships. With these, they can compare interactions across all sites as well as specific sites or species. This analysis pulls out a relatively strong negative effect of soil warming on AMF colonization, as well as number of indirect effects that vary from positive to negative. This suggests that patterns of response to warming mediated by AMF may in part be challenging to predict across different species and sites, but are likely to lead to a decline in AMF rate, and possible declines in biomass of many species.
The manuscript clearly frames the question and identifies a gap in the literature (multi-site investigations of warming on AMF colonization), is done to a high technical and statistical standard, is sufficiently described that another researcher should be able to replicate the work, and adheres to prevailing ethical standards.

Validity of the findings

This experiment shows the challenges, and the importance, of large, multisite experiments. Although the patterns are not clear, it shows that warming is going to have a multitude of direct and indirect effects, many of which may be site, year, or species specific.
I feel the data will be publicly available, that the approaches are statistically robust (admitting that SEMs can be quirk), that the conclusions drawn from them are justified and without excessive speculation.

Additional comments

Although generally well written, there are some important typos in the manuscript. I have detailed those I caught below. However, I almost certainly missed some.
Line 37: in the first first sentence of the introduction, misspells "symbionts."
Line 40: the awkward phrasing of "terrestrial C and nutrient cycles" which I would recommend spelling out as carbon.
Line 68: Mycorrhizal "measures" is an awkward term. Is there a more elegant way to say this?
Line 100: There is an extra word in this sentence- delete ”exist”

---

## Round 0.2 · accepted · Accept

The authors have addressed satisfactorily the concerns and suggestions by the reviewers.

Reviewer 3 ·

Basic reporting

The authors have addressed all of my concerns. The other reviewer had a greater number of concerns, which also have all been mentioned.

Experimental design

No concerns have been raised about the design.

Validity of the findings

No concerns have been raised about the findings.

Additional comments

Looking forward to seeing this in print.